# Neural Surface Reconstruction of Dynamic Scenes with Monocular RGB-D Camera

**Hongrui Cai[†], Wanquan Feng[†], Xuetao Feng[‡], Yan Wang[‡], Juyong Zhang[†][*]**

[†]University of Science and Technology of China  [‡]Alibaba Group

https://ustc3dv.github.io/ndr

## Abstract

We propose Neural-DynamicReconstruction (NDR), a template-free method to recover high-fidelity geometry and motions of a dynamic scene from a monocular RGB-D camera. In NDR, we adopt the neural implicit function for surface representation and rendering such that the captured color and depth can be fully utilized to jointly optimize the surface and deformations. To represent and constrain the non-rigid deformations, we propose a novel neural invertible deforming network such that the cycle consistency between arbitrary two frames is automatically satisfied. Considering that the surface topology of dynamic scene might change over time, we employ a topology-aware strategy to construct the topology-variant correspondence for the fused frames. NDR also further refines the camera poses in a global optimization manner. Experiments on public datasets and our collected dataset demonstrate that NDR outperforms existing monocular dynamic reconstruction methods.

## 1  Introduction

Reconstructing 3D geometry shape, texture and motions of the dynamic scene from a monocular video is a classical and challenging problem in computer vision. It has broad applications in many areas like virtual and augmented reality. Although existing methods [63, 65] have demonstrated impressive reconstruction results for dynamic scenes only with 2D images, they are still difficult to recover high-fidelity geometry shapes, especially for some casually captured data as abundant potential solutions exist without depth constraints. Only with 2D measurements, dynamic reconstruction methods require that motions of interested object hold in a nearby $z$-plane. Meanwhile, it is difficult to construct reliable correspondences in areas with weak texture, which causes error accumulation in the canonical space.

To solve this under-constrained problem, some methods propose to utilize shape priors for some special object types. For example, category-specific parametric shape models like 3DMM [6], SMPL [41] and SMAL [72] are first constructed and then used to help the reconstruction. However, templated-based methods could not generalize to unknown object types. On the other hand, some methods utilize annotations, like keypoints and optical flow, obtained from manual annotators or off-the-shelf tools [31, 33, 63, 65]. The motion trajectories of sparse or dense 2D points can effectively help recover the exact motion of the whole structure. However, it needs human labeling for supervision or highly depends on the quality of learned priors from a large-scale dataset.

One straightforward solution to this under-constrained problem is to reconstruct the interested object based on observations from RGB-D cameras like Microsoft Kinect [69] and Apple iPhone X. Existing fusion-based methods [44, 27, 54] utilize a dense non-rigid warp field and a canonical truncated signed

---

[*]Corresponding author.

36th Conference on Neural Information Processing Systems (NeurIPS 2022).

distance (TSDF) volume to represent motion and shape, respectively. However, these fusion-based methods might fail due to accumulated tracking errors, especially for long sequences. To alleviate this problem, some learning-based methods [9, 8, 39] utilize more accurate correspondences which are annotated or learned from synthesis datasets to guide the dynamic fusion process. However, the captured color and depth information is not represented together within one differentiable framework in these methods. Recently, a neural implicit representation based method [3] has been proposed to reconstruct a room-scale scene from RGB-D inputs, but it is only designed for static scenes and can not be directly applied to dynamic scenes.

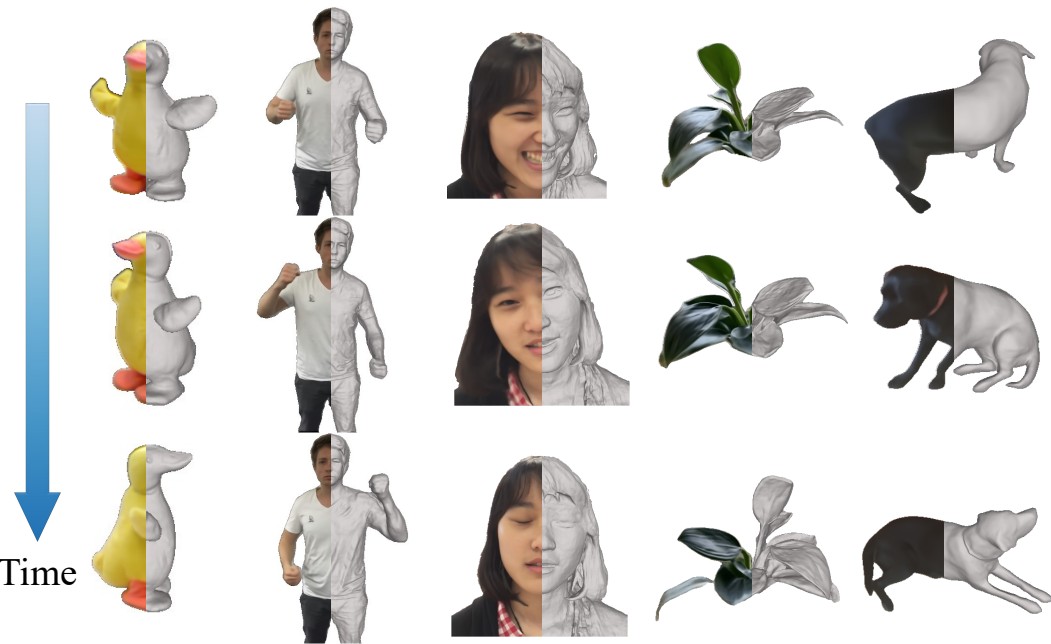

Figure 1: Examples of *reconstructed* (right) and *rendered* (left) results by NDR. Given a monocular RGB-D video sequence, NDR recovers high-fidelity geometry and motions of a dynamic scene.

In this paper, we present Neural-DynamicReconstruction (**NDR**), a neural dynamic reconstruction method from a monocular RGB-D camera (Fig. 1). To represent the high-fidelity geometry and texture of deformable object, NDR maintains a neural implicit field as the canonical space. With extra depth constraint, there still exist multiple potential solutions since the correspondences between different frames are still unknown. In this paper, we propose the following strategies to constrain and regularize the solution space: (1) integrating all RGB-D frames to a high-fidelity textured shape in the canonical space; (2) maintaining cycle consistency between arbitrary two frames; (3) a surface representation which can handle topological changes.

Specifically, we adopt the neural SDF and radiance field to respectively represent the high-fidelity geometry and appearance in the canonical space instead of the TSDF volume frequently used in fusion-based methods [28, 44, 54, 9, 8, 39]. In our framework, each RGB-D frame can be integrated into the canonical representation. We propose a novel neural deformation representation that implies a continuous bijective map between observation and canonical space. The designed invertible module applies a cycle consistency constraint through the whole RGB-D video; meanwhile, it fits the natural properties of non-rigid motion well. To support topology changes of dynamic scene, we adopt the topology-aware network in HyperNeRF [47]. Thanks for modeling topology-variant correspondence, our framework can handle topology changes while existing deformation graph based methods [44, 39, 65] could not. NDR also further refines camera intrinsic parameters and poses during training. Extensive experimental results demonstrate that NDR can recover high-fidelity geometry and photorealistic texture for monocular category-agnostic RGB-D videos.

## 2 Related Works

**RGB based dynamic reconstruction.**    Dynamic reconstruction approach can be divided into template-based and template-free types. Templates [6, 41, 50, 72] are category-specific statistical models constructed from large-scale datasets. With the help of pre-constructed 3D morphable models [6, 12, 36], some researches [5, 11, 26, 57, 21, 25, 19] reconstruct faces or heads from RGB inputs. Most of them need 2D keypoints as extra supervisory information to guide dynamic tracking [71, 17, 19]. With the aid of human parametric models [1, 41], some works [7, 62, 23, 24, 70, 29] recover digital avatars based on monocular image or video cues. However, it is unpractical to extend templates to general objects with limited 3D scanned priors, such as articulated objects, clothed human and animals. Non-rigid structure from motion (NR-SFM) algorithms [10, 51, 15, 34, 53] are to reconstruct category-agnostic object from 2D observations. Although NR-SFM can reconstruct reasonable result for general dynamic scenes, it heavily depends on reliable point trajectories throughout observed sequences [52, 56]. Recently, some methods [63, 64, 65] obtain promising results from a long monocular video or several short videos of a category. LASR [63] and ViSER [64] recover articulated shapes via a differentiable rendering manner [40], while BANMo [65] models them with the help of Neural Radiance Fields (NeRF) [43]. However, due to the depth ambiguity of input 2D images, the reconstruction might fails for some challenging inputs.

**RGB-D based dynamic reconstruction.**  Recovering 3D deforming shapes from a monocular RGB video is a highly under-constrained problem. On the other hand, The progress in consumer-grade RGB-D sensors has made depth map capture from a single camera more convenient. Therefore, it is quite natural to reconstruct the target objects based on RGB-D sequences. DynamicFusion [44], the seminar work of RGB-D camera based dynamic object reconstruction, proposes to estimate a template-free 6D motion field to warp live frames into a TSDF surface. The surface representation strategy has also been used in KinectFusion [28]. VolumeDeform [27] represents motion in a grid and incorporates global sparse SIFT [42] features during alignment. Guo et al. [20] coheres albedo, geometry and motion estimation in an optimization pipeline. KillingFusion [54] and SobolevFusion [55] are proposed to deal with topology changes. During deep learning era, DeepDeform [9] and Bozic et al. [8] aim to learn more accurate correspondences for tracking improvement of faster and more complex motions. OcclusionFusion [39] probes and handles the occlusion problem via an LSTM-involved graph neural network but fails when topology changes. Although these methods obtain promising reconstruction results with the additional depth cues, their reconstructed shapes mainly depend on the captured depths, while the RGB images are not fully utilized to further improve the results.

**Dynamic NeRF.**  Given a range of image cues, prior works on NeRF [43] optimize an underlying continuous scene function for novel view synthesis. Some NeRF-like methods [37, 49, 58, 18, 46, 47] achieve promising results on dynamic scenes without prior templates. Nerfies [46] and AD-NeRF [22] reconstruct free-viewpoint selfies from monocular videos. HyperNeRF [47] models an ambient slicing surface to express topologically varying regions. Recent approaches [60, 3] introduce neural representation for static object/scene reconstruction, but theirs can not be used for non-rigid scenes.

**Cycle consistency constraint.**    To maintain cycle consistency between deformed frames is an important regularization in perceiving and modeling dynamic scenes [61]. However, recent methods [64, 37, 65] try to leverage a loss term to constrain estimated surface features or scene flow, which is a weak but not strict property. Therefore, constructing an invertible representation for deformation field is a reasonable design. Several invertible networks are proposed to represent deformation, such as Real-NVP [16], Neural-ODE [13], I-ResNet [4]. Based on these manners, there exist some methods modeling deformation in space [30, 66, 48] or time [45, 35] domain. CaDeX [35] is a novel dynamic surface representation method using a real-valued non-volume preserving module [16]. Different from these strategies, we propose a novel scale-invariant binary map between observation space and 3D canonical space to process RGB-D sequences, which is more suitable for modeling non-rigid motion.

## 3 Method

The input of NDR is an RGB-D sequence $\{(\mathcal{I}_i, \mathcal{D}_i), i = 1, \cdots, N\}$ captured by a monocular RGB-D camera (e.g., Kinect and iPhone X), where $\mathcal{I}_i \in \mathbb{R}^{H \times W \times 3}$ is the $i$-th RGB frame and

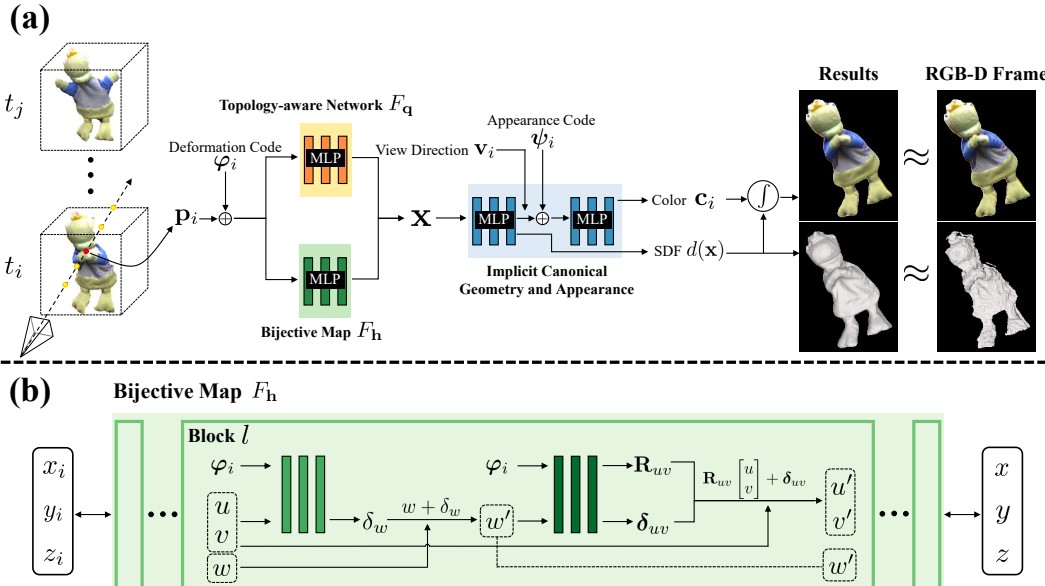

Figure 2: (a) Pipeline of NDR. (b) The structure of bijective map $F_{\mathbf{h}}$ in the deformation field.

$\mathcal{D}_i \in \mathbb{R}^{H \times W \times 1}$ is the corresponding aligned depth map. To optimize a canonical textured shape and motion through the sequence, we leverage full $N$ color frames $\mathcal{I}_i$ as well as corresponding depth frames $\mathcal{D}_i$. Specifically, we first adopt video segmentation methods [14, 38] to obtain the mask $\mathcal{M}_i$ of interested object. Then, we integrate RGB-D video sequence into a canonical hyper-space composed of a 3D canonical space and a topology space. We propose a continuous bijective representation between the 3D canonical and observation space such that the cycle consistency can be strictly satisfied. The implicit surface is represented by a neural SDF and volume rendering field, as a function of input hyper-coordinate and camera view. The geometry, appearance, and motions of dynamic object are optimized without any template or structured priors, like optical flow [65], 2D annotations [9] and estimated normal map [29]. The pipeline of NDR is shown in Fig. 2(a).

## 3.1 Bijective Map in Space-time Synthesis

**Invertible representation.** Given a 3D point sampled in the space of $i$-th frame, recent methods [44, 65, 47] model its motion as a 6D transformation in $\mathbf{SE}(3)$ space. Nerfies [46] and Hyper-NeRF [47] construct a continuous dense field to estimate the motion. To reduce the complexity, DynamicFusion [44] and BANMo [65] define warp functions based on several control points. The latter designs both 2D and 3D cycle consistency loss terms to apply bijective constraints to deformation representation, but it is just a guide for learning instead of a rigorous inference module. Similar to the previous works, we also construct the deformation between each current frame and the 3D canonical space. Further, we employ a strictly invertible bijective mapping, which is naturally compatible with the cycle consistency strategy. Specifically, we decompose the non-rigid deformation into several reversible bijective blocks, where each block represents the transformation along and around a certain axis. In this manner, our deformation representation is strictly invertible and fits the natural properties of non-rigid motion well, which is helpful for the reconstruction effect.

We denote $\mathbf{p}_i = [x_i, y_i, z_i] \in \mathbb{R}^3$ as a position of the observation space at time $t_i$, in which a deformed surface $U_i$ is embedded. It is noticeable that $\mathbf{p}_i$ represents any position, both surface and free-space points. A continuous homeomorphic mapping $\mathcal{H}_i : \mathbb{R}^3 \to \mathbb{R}^3$ maps $\mathbf{p}_i$ back to the 3D canonical position $\mathbf{p} = [x, y, z]$. Supposes that there exists a canonical shape $U$ of the interested object, which is independent of time and is shared across the video sequence. Notes that map $\mathcal{H}_i$ is invertible, and thus we can directly obtain the deformed surface at time $t_i$:

$$U_i = \{\mathcal{H}_i^{-1}([x, y, z]) | \forall [x, y, z] \in U\}. \tag{1}$$

Then, the correspondence of $\mathbf{p}_i$ can be expressed by the bijective map, factorized as:

$$[x_j, y_j, z_j] = \mathcal{G}_{ij}([x_i, y_i, z_i]) = \mathcal{H}_j^{-1} \circ \mathcal{H}_i([x_i, y_i, z_i]). \tag{2}$$

The deformation representation $\mathcal{G}$ is cycle consistent strictly, since it is invariant on deforming path ($\mathcal{G}_{jk} \circ \mathcal{G}_{ij} = \mathcal{G}_{ik}$). As a composite function of two bijective maps (Eq. 2), it is a topology-invariant function between arbitrary double time stamps.

**Implementation.** Based on these observations, we implement the bijective map $\mathcal{H}$ by a novel invertible network $h$. While Real-NVP [16] seems a suitable network structure, its scale-variant property limits its usage in our object reconstruction task. Inspired by the idea of Real-NVP to split the coordinates, we decompose our scale-invariant deformation into several blocks. In each block, we set an axis and represent the motion steps as simple axis-related rotations and translations, which are totally shared by the forward and backward deformations. In this manner, the inverse deformation $\mathcal{H}^{-1}$ can be viewed as the composite of the inverse of these simple rotations and translations in $\mathcal{H}$. On the other hand, this map also regularizes the freedom of deformation.

Fig. 2(b) shows the detailed structure of each block. Given a latent deformation code $\boldsymbol{\varphi}$ binding with time, we firstly consider the forward deformation, where the 3D positions $[u, v, w] \in \mathbb{R}^3$ of observation space is input, and the positions $[u', v', w'] \in \mathbb{R}^3$ of 3D canonical space is output. The cause of the invertible property is that after specifying a certain coordinate axis, each block predicts the movement along and rotation around the axis in turn, and the process of predicting the deformation is reversible, owing to coordinate split. In the inverse process, each block can infer the rotation around and movement along the axis from $[u', v', w']$ and invert them in turn to recover the original $[u, v, w]$.

Without loss of generality, let the $w$-axis to be the chosen axis. With $[u, v]$ fixed, we compute a displacement $\delta_w$ and update $w'$ as $w + \delta_w$. With $[w']$ fixed, we then compute the rotation $\mathbf{R}_{uv}$ and translation $\boldsymbol{\delta}_{uv}$ for $[u, v]$ and update them as $[u', v']$. Oppositely, for the backward deformation, we apply $-\boldsymbol{\delta}_{uv}$, $\mathbf{R}_{uv}^{-1}$, and $-\delta_w$ in turn to recover $[u', v', w']$ back to $[u, v, w]$. We refer the reader to supplementary material for the inverse process. Therefore, if the network $\mathbf{h}$ consists of these invertible blocks, it can represent a bijective map as well. At time $t_i$, $\mathbf{h}(\cdot|\boldsymbol{\varphi}_i) : \mathbb{R}^3 \rightarrow \mathbb{R}^3$ maps 3D positions $\mathbf{p}_i$ of observation space back to 3D canonical correspondences $\mathbf{p}$, where $\boldsymbol{\varphi}_i$ denotes the deformation code of $i$-th frame. In our experiment, we use a Multi-Layer Perceptron (MLP) as the implementation of $\mathbf{h}$, so we design a continuous bijective map $F_{\mathbf{h}}$ for space-time synthesis.

## 3.2 Deformation Field

Although the proposed deformation representation is a continuous homeomorphic mapping that satisfies the cycle consistency between different frames, it also preserves the surface topology. However, several dynamic scenes (e.g., varying body motion and facial expression) may undergo topology changes. Therefore, we combine a topology-aware design [47] into our deformation field. 3D positions $\mathbf{p}_i$ observed at time $t_i$ are mapped to topology coordinates $\mathbf{q}(\mathbf{p}_i)$ through a network $\mathbf{q} : \mathbb{R}^3 \rightarrow \mathbb{R}^m$. We regress topology coordinates from an MLP $F_{\mathbf{q}}$. Then the corresponding coordinate of $\mathbf{p}_i$ in the canonical hyper-space is represented as:

$$\mathbf{x} = [\mathbf{p}, \mathbf{q}(\mathbf{p}_i)] = [F_{\mathbf{h}}(\mathbf{p}_i, \boldsymbol{\varphi}_i), F_{\mathbf{q}}(\mathbf{p}_i, \boldsymbol{\varphi}_i)] \in \mathbb{R}^{3+m}, \tag{3}$$

conditioned on time-varying deformation $\boldsymbol{\varphi}_i$.

## 3.3 Implicit Canonical Geometry and Appearance

Inspired by NeRF [43], we consider that a sample point $\mathbf{x} \in \mathbb{R}^{3+m}$ in the canonical hyper-space is associated with two properties: density $\sigma$ and color $\mathbf{c} \in \mathbb{R}^3$.

**Neural SDF.** Notes that the object embeds in the $(3 + m)$-D canonical hyper-space. In this work, we represent its geometry as the zero-level set of an SDF:

$$S = \{\mathbf{x} \in \mathbb{R}^{3+m} | d(\mathbf{x}) = 0\}. \tag{4}$$

Following NeuS [60], we utilize a probability function to calculate the density value $\sigma(\mathbf{x})$ based on the estimated signed distance value, which is an unbiased and occlusion-aware approximation. We refer the reader to their paper for more details.

**Implicit rendering network.** We utilize a neural renderer $F_{\mathbf{c}}$ as the implicit appearance network. At time $t_i$, it takes in a 3D canonical coordinate $\mathbf{p}$, its corresponding normal, a canonical view direction as well as a geometry feature vector, then outputs the color of the point, conditioned on a time-varying appearance code $\psi_i$. Specifically, we first compute its normal $\mathbf{n_p} = \nabla_{\mathbf{p}}d(\mathbf{x})$ by gradient calculation. Then, the view direction $\mathbf{v_p}$ in 3D canonical space can be obtained by transforming the view direction $\mathbf{v}_i$ in observation space with the Jacobian matrix $J_{\mathbf{p}}(\mathbf{p}_i) = \partial\mathbf{p}/\partial\mathbf{p}_i$ of the 3D canonical map $\mathbf{p}$ w.r.t $\mathbf{p}_i$: $\mathbf{v_p} = J_{\mathbf{p}}(\mathbf{p}_i)\mathbf{v}_i$. Except the SDF value, we adopt a larger MLP $F_d(\mathbf{x}) = (d(\mathbf{x}), \mathbf{z}(\mathbf{x}))$ to compute the embedded geometry feature $\mathbf{z_x} = \mathbf{z}(\mathbf{x})$ to help the prediction of global shadow [67]. Finally, noticing $\mathbf{p}_i$ is the correspondence of $\mathbf{x}$ at time $t_i$, we can formulate its color $\mathbf{c}_i$ as:

$$\mathbf{c}_i = F_{\mathbf{c}}(\mathbf{p}, \mathbf{n_p}, \mathbf{v_p}, \mathbf{z_x}, \psi_i) = F_{\mathbf{c}}(\mathbf{p}, \nabla_{\mathbf{p}}d(\mathbf{x}), J_{\mathbf{p}}(\mathbf{p}_i)\mathbf{v}_i, \mathbf{z}(\mathbf{x}), \psi_i). \tag{5}$$

It can be seen that the color of point $\mathbf{p}_i$ viewed from direction $\mathbf{v}_i$ depends on the deformation field, canonical representation, a deformation code as well an appearance code combined with time.

### 3.4 Optimization

Given an RGB-D sequence with the masks of interested object $\{(\mathcal{I}_i, \mathcal{D}_i, \mathcal{M}_i), i = 1, 2, \cdots, N\}$, the optimizable parameters include MLPs $\{F_{\mathbf{h}}, F_{\mathbf{q}}, F_d, F_{\mathbf{c}}\}$, learnable codes $\{\varphi_i, \psi_i\}$, RGB and depth camera intrinsics $\{\mathcal{K}_{\text{rgb}}, \mathcal{K}_{\text{depth}}\}$, as well as $\mathbf{SE}(3)$ camera pose $\mathcal{T}_i$ at each time $t_i$. Our target is to design the loss terms to match input masks, color images and depth images. Since we leverage neural implicit functions for representing the geometry, appearance and motion of dynamic object, we divide all constraints into two parts, on free-space points and on surface points:

$$\mathcal{L} = \underbrace{\left(\lambda_1 \mathcal{L}_{\text{mask}} + \lambda_2 \mathcal{L}_{\text{color}} + \lambda_3 \mathcal{L}_{\text{depth}} + \lambda_4 \mathcal{L}_{\text{reg}}\right)}_{\text{free-space}} + \underbrace{\left(\lambda_5 \mathcal{L}_{\text{sdf}} + \lambda_6 \mathcal{L}_{\text{visible}}\right)}_{\text{surface}}, \tag{6}$$

where $\lambda_j (j = 1, 2, \cdots, 6)$ are balancing weights.

**Constraints on free-space.** Given a ray parameterized as $\mathbf{r}(s) = \mathbf{o} + s\mathbf{v}$ (pass through a pixel), we sample the implicit radiance field at points lying along this ray to approximate its color and depth:

$$\hat{\mathbf{C}}(\mathbf{r}) = \int_{s_n}^{s_f} T(s)\sigma(s)\mathbf{c}(s)\,\mathrm{d}s, \quad \hat{\mathbf{D}}(\mathbf{r}) = \int_{s_n}^{s_f} T(s)\sigma(s)s\,\mathrm{d}s, \tag{7}$$

where $s_n$ and $s_f$ represent near and far bounds, and $T(s) = \exp(-\int_{s_n}^{s} \sigma(u)\,\mathrm{d}u)$ denotes the accumulated transmittance along the ray. The density and color calculation are described in Sec. 3.3. Then the color and depth reconstruction loss are defined as:

$$\mathcal{L}_{\text{color}} = \sum_{\mathbf{r} \in \mathcal{R}(\mathcal{K}_{\text{rgb}}, \mathcal{T}_i)} \|M(\mathbf{r})(\hat{\mathbf{C}}(\mathbf{r}) - \mathbf{C}(\mathbf{r}))\|_1, \tag{8}$$

$$\mathcal{L}_{\text{depth}} = \sum_{\mathbf{r} \in \mathcal{R}(\mathcal{K}_{\text{depth}}, \mathcal{T}_i)} \|M(\mathbf{r})(\hat{\mathbf{D}}(\mathbf{r}) - \mathbf{D}(\mathbf{r}))\|_1, \tag{9}$$

where $\mathcal{R}(\mathcal{K}_{\text{rgb}}, \mathcal{T}_i)$ and $\mathcal{R}(\mathcal{K}_{\text{depth}}, \mathcal{T}_i)$ represent the set of rays to RGB and depth camera, respectively. $M(\mathbf{r}) \in \{0, 1\}$ is the object mask value, while $\mathbf{C}(\mathbf{r})$ and $\mathbf{D}(\mathbf{r})$ are the observed color and depth value. To focus on dynamic object reconstruction, we also define a mask loss as

$$\mathcal{L}_{\text{mask}} = \text{BCE}(\hat{M}(\mathbf{r}), M(\mathbf{r})), \tag{10}$$

where $\hat{M}(\mathbf{r}) = \int_{s_n}^{s_f} T(s)\sigma(s)\,\mathrm{d}s$ is the density accumulation along the ray, and BCE is the binary cross entropy loss.

An Eikonal loss is introduced to regularize $d(\mathbf{x})$ to be a signed distance function of $\mathbf{p}$, and it has the following form:

$$\mathcal{L}_{\text{reg}} = \sum_{\mathbf{x} \in \mathcal{X}} (\|\nabla_{\mathbf{p}}d(\mathbf{x})\|_2 - 1)^2, \tag{11}$$

where $\mathbf{x}$ are points sampled in the canonical hyper-space $\mathcal{X}$. In our implementation, to obtain $\mathbf{x}$, we first sample some points $\mathbf{p}_i$ on the observed free-space and then deform sampled points back to $\mathcal{X}$ using Eq. 3. We constrain points sampled by a uniform and importance sampling strategy.

**Constraints on surface.** Except for the losses on the free-space, we also constrain the property of points lying on the depth images $\mathcal{D}_i$. We add an SDF loss term:

$$\mathcal{L}_{\mathrm{sdf}} = \sum_{\mathbf{p}_i \in \mathcal{D}_i} \|d(\mathbf{x})\|_1. \tag{12}$$

To avoid the deformed surface at each time fuses into the canonical space which causes multi-surfaces phenomenon, we design a visible loss term to constrain surface:

$$\mathcal{L}_{\mathrm{visible}} = \sum_{\mathbf{p}_i \in \mathcal{D}_i} \max(\langle \frac{\mathbf{n_p}}{\|\mathbf{n_p}\|_2}, \frac{\mathbf{v_p}}{\|\mathbf{v_p}\|_2} \rangle, 0), \tag{13}$$

where $\langle \cdot, \cdot \rangle$ denotes the inner product. The visible loss term is to constrain the angle between the normal vector of the sampled point on depth map and the view direction to be larger than 90 degrees, which aims to guide depth points to be visible surface points under the RGB-D camera view.

## 4 Experiments

### 4.1 Experimental Settings

**Implementation details.** We initialize $d(\mathbf{x})$ such that it approximates a unit sphere [2]. We train our neural networks using the ADAM optimizer [32] with a learning rate $5 \times 10^{-4}$. We run most of our experiments with $6 \times 10^4$ iterations for 12 hours on a single NVIDIA A100 40GB GPU. On free-space, we sample $2,048$ rays per batch (128 points along each ray). Following NeuS [60], we first uniformly sample 64 points, and then adopt importance sampling iteratively for 4 times (16 points each iteration). On depth map, we uniformly sample $2,048$ points per batch. For coarse-to-fine training, we utilize an incremental positional encoding strategy on sampled points, similar with Nerfies [46]. The weights in Eq. 6 are set as: $\lambda_1 = 0.1, \lambda_2 = 1.0, \lambda_3 = 0.5, \lambda_4 = 0.1, \lambda_5 = 0.5, \lambda_6 = 0.1$.

For non-rigid object segmentation, we leverage off-the-shelf methods, RVM [38] for human and MiVOS [14] for other objects. Since we assume the region of object is inside a unit sphere, we normalize the points back-projected from depth maps first. If the collected sequence implies larger global rotation, we leverage Robust ICP method [68] for per-frame initialization of poses $\mathcal{T}_i$.

**Datasets.** To evaluate our NDR and baseline approaches, we use 6 scenes from DeepDeform [9] dataset, 7 scenes from KillingFusion [54] dataset, 1 scene from AMA [59] dataset and 11 scenes captured by ourselves. The evaluation data contains 6 classes: human faces, human bodies, domestic animals, plants, toys, and clothes. It includes challenging cases, such as rapid movement, self-rotation motion, topology change and complex shape. DeepDeform [9] dataset is captured by an iPad. Its RGB-D streams are recorded and aligned at a resolution of $640 \times 480$ and 30 frames per second. Since our NDR does not need any annotated or estimated correspondences, we only leverage RGB-D sequences and camera intrinsics as initialization when evaluating NDR, without scene flow or optical flow data. We choose 6 scenes from the whole dataset, including human bodies, dogs, and clothes. All sequences in KillingFusion [54] dataset were recorded with a Kinect v1, also aligned to $640 \times 480$ resolution. We choose all scenes from it, which contain toys and human motions. For evaluation on synthetic data, we use AMA [59] dataset, which contains reconstructed mesh corresponding to each video frame. To construct synthetic depth data, we render meshes to a chosen camera view. In the experiment, we do not utilize any multi-view messages but only monocular RGB-D frames. To increase the data diversity, especially for adding more challenging but routine conditions (e.g., topology change and complex details), we capture some human head and plant videos with iPhone X (resolution $480 \times 640$ at 30 fps). When capturing head data, we ask the person to rotate the face while freely varying expressions. When capturing plant data, we record the states of leaf swings.

**Comparison methods.** (1) A widely-used classical fusion-based method, DynamicFusion [44]: It is the pioneering work that estimates and utilizes the motion of hierarchical node graph for deforming guidance, and it assumes the shape inside a canonical TSDF volume. (2) Two recent fusion-based methods, DeepDeform [9] and Bozic et al. [8]: These methods utilize the learning-based correspondences to help handle challenging motions. (3) A state-of-the-art fusion-based method, OcclusionFusion [39]: It computes occlusion-aware 3D motion through a neural network for modeling guidance. (4) A state-of-the-art RGB reconstruction method from monocular video, BANMo [65]: It models articulated 3D shapes in a neural blend skinning and differentiable rendering framework. For

comparison with RGB-D based methods, we use our re-implementation of DynamicFusion [44] and the results provided by the authors of OcclusionFusion [39].

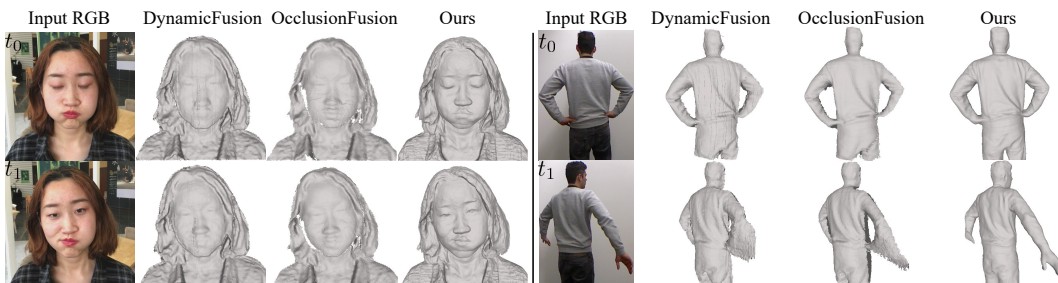

Figure 3: Qualitative comparisons with DynamicFusion [44] and OcclusionFusion [39] on our dataset (a sequence of 600 frames) and KillingFusion [54] dataset (a sequence of 200 frames).

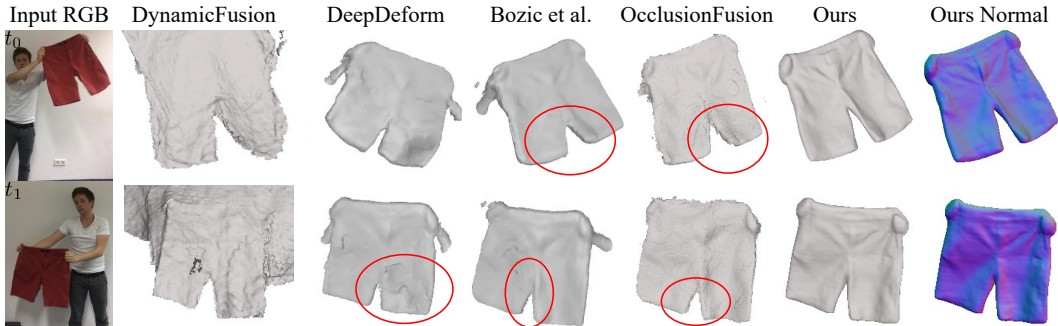

Figure 4: Qualitative comparisons with DynamicFusion [44], DeepDeform [9], Bozic et al. [8] and OcclusionFusion [39] on DeepDeform [9] dataset (a sequence of 500 frames). The results of DeepDeform [9] and Bozic et al. [8] are copied from the video of Bozic et al. [8].

## 4.2 Comparisons

**RGB-D based methods.** For qualitative evaluation, we exhibit some comparisons with Dynamic-Fusion [44] and OcclusionFusion [39] in Fig. 3, also with DeepDeform [9] and Bozic et al. [8] in Fig. 4. Specifically, results of detailed modeling verify that bijective deformation mapping help match photometric correspondences between observed frames. As Fig. 3 shown, our NDR models geometry details while fusion-based methods [44, 39] are easy to form artifacts on the reconstructed surfaces. NDR also achieves considerable reconstruction accuracy on handling rapid movement (Fig. 4).

For quantitative evaluation, we calculate geometry errors on some testing sequences, following previous works [9, 8, 39]. The geometry metric is to compare depth values inside the object mask to the reconstructed geometry. The sequences are on behalf of various class objects and cases, including domestic animal (seq. *Dog* from DeepDeform [9]), rotated body,

| Metric | Method | Dataset | | | | | |
|---|---|---|---|---|---|---|---|
| | | DeepDeform | KillingFusion | | | Human Head | |
| | | Dog | Alex | Hat | Frog | Human1 | Human2 |
| Mean | DynamicFusion | 92.53 | 5.39 | 9.92 | 2.75 | 1.87 | 1.63 |
| | OcclusionFusion | 3.64 | **3.75** | 6.77 | 1.61 | 1.30 | 1.25 |
| | Ours | **3.42** | 4.24 | **4.93** | **1.34** | **1.26** | **1.08** |
| Median | DynamicFusion | 12.25 | 5.19 | 6.72 | 2.74 | 1.67 | 1.59 |
| | OcclusionFusion | 3.48 | **3.41** | 6.47 | 1.59 | 1.30 | 1.25 |
| | Ours | **3.39** | 4.11 | **4.66** | **1.33** | **1.25** | **1.07** |

Table 1: Quantitative results on 6 sequences. The geometry error represents the difference between reconstructed shape and depth values inside the mask. All values are in *mm*.

human-object interaction, general object (seq. *Alex, Hat, Frog* from KillingFusion [54], separately), and human heads (seq. *Human1, Human2* from our collected dataset). The quantitative results are shown in Tab. 1. We can see that our NDR outperforms previous works [44, 39], owing to jointly optimizing geometry, appearance and motion on a total video. On seq. *Alex*, the geometry error of OccluionFusion [39] is lower than that of ours. However, NDR can handle topology varying well, as shown in the corresponding qualitative results on the right of Fig. 3.

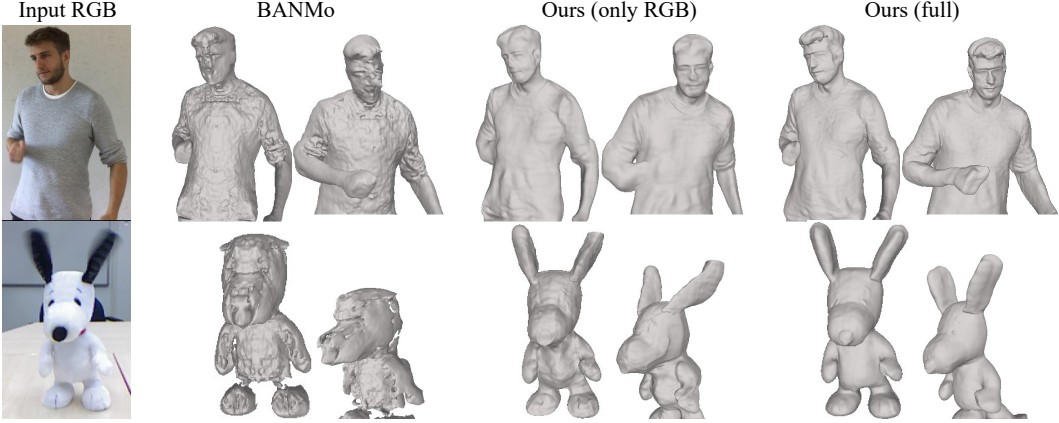

Figure 5: Qualitative comparison with BANMo [65] and ablation study on optimization without depth images on DeepDeform [9] dataset and KillingFusion [54] dataset (both sequences of 200 frames). Each sample is shown in two view directions (the camera view and a novel view).

**RGB based method.** Fig. 5 exhibits several comparisons with a recent RGB based method - BANMo [65]. BANMo takes the RGB sequence as input and optimizes the geometry, appearance and motion based on the precomputed annotations, including the camera pose and optical flow. For a fair comparison, we also compare BANMo [65] with our NDR with only RGB supervision, where we provide them with the same camera initialization and frame-wise mask. For the RGB-only situation, both our method and BANMo may make some structural mistakes, such as the human arm in ours and the Snoopy's ears in BANMo. Moreover, compared to our RGB-only results, BANMo suffers more from the local geometry noise, which should be due to the error caused by incorrect precomputed annotations. Meanwhile, our method does not rely on any precomputed annotations and achieves flat results. With the RGB-D sequence as input, our NDR full model performs robust and well in modeling geometry details and rapid motions.

## 4.3 Robustness on Camera Initialization

In order to systematically analyze the performance of our camera pose optimization ability, we add an experiment to test the robustness under various degrees of noise on both real and synthetic data. We choose 2 sequences of small rigid motion separately from DeepDeform [9] dataset (a body with moving joints, 200 frames) and AMA [59] dataset (a Samba dancer, 175

|  | 0 | 5 | 10 | 20 | 40 | 60 |
|---|---|---|---|---|---|---|
| Moving Joints | 2.95 | 4.58 | 6.58 | 9.17 | 11.07 | 30.28 |
| Samba Dancer | 3.90 | 5.49 | 8.18 | 11.60 | 13.46 | 27.02 |

Table 2: Robustness on camera pose initialization. Values of the first row represent degrees of added Gaussian noises. Other values are mean geometry errors on the sequences, in *mm*.

monocular frames). As Tab. 2, we add Gaussian noises with $5, 10, 20, 40, 60$ degrees of standard deviation to initial Euler angles and calculate mean geometry errors (0 denotes without adding noises). The results show that NDR is robust against noisy camera poses to a certain extent, owing to its neural implicit representation and abundant optimization with RGB-D messages. If the standard deviation of Gaussian Noises is over 20 degrees, the reconstruction quality will be obviously affected (geometry error is over 1 *cm*). We refer the reader to supplementary material for qualitative results.

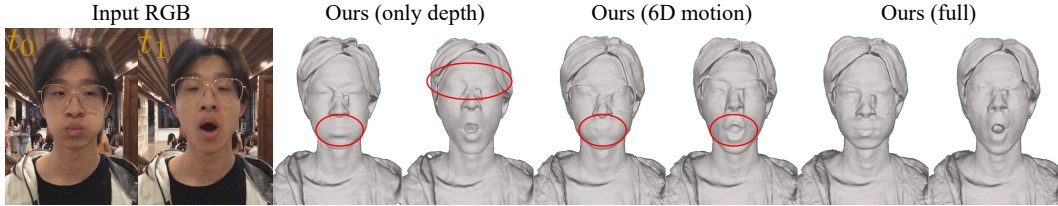

Figure 6: Ablation studies on optimization without RGB images and using 6D motion instead of the bijective map in the deformation field. The example sequence contains 200 frames.

## 4.4 Ablation Studies

We evaluate 3 components of our NDR regarding their effects on the final reconstruction result.

**Depth cues.** We evaluate the reconstruction results with only RGB supervision, i.e. removing depth images and only supervised with loss terms $\mathcal{L}_{mask}, \mathcal{L}_{color}, \mathcal{L}_{reg}$. As shown in Fig. 5, the reconstruction results with only RGB information are not correct (especially seen from a novel view) since monocular camera scenes exist the ambiguity of depth.

**RGB cues.** We also evaluate the reconstruction results with only depth supervision, i.e. removing RGB images and color loss term $\mathcal{L}_{color}$. As shown in Fig. 6, the reconstructed shapes lack geometrical details as color messages are not used.

**Bijective map $F_{\mathbf{h}}$.** To verify the effect of our proposed bijective map $F_{\mathbf{h}}$ (Sec. 3.1), we change it to 6D motion representation in $\mathbf{SE}(3)$ space. As shown in Fig. 6, since $F_{\mathbf{h}}$ can satisfy the cycle consistency strictly, it is less prone to accumulate artifacts and thus performs better in local geometry. In comparison, the irreversible transformation is easy to fail in preserving high-quality surfaces.

## 4.5 Evaluation of Cycle Consistency

We perform a numerical experiment for cycle consistency evaluation on the whole deformation field. In the experiment, we randomly select 3 frames (indexed by $i, j, k$) as a group in a video sequence. Given points on one frame, we calculate the corresponding coordinates on another frame and record this scene flow as $\mathbf{f}$. Then it includes 2 deforming paths from frame $i$ to $k$, based on the direct flow $\mathbf{f}_{ik}$, or the composite flow $\mathbf{f}_{ij} + \mathbf{f}_{jk}$. To evaluate the

|  | w/ $F_{\mathbf{h}}$ | w/ 6D motion |
|---|---|---|
| Rotated Body | **5.29** | 473.25 |
| Talking Head | **4.97** | 289.41 |

Table 3: Evaluation of cycle consistency. Each value is the mean error ($\times 10^{-4}$) per point in the unit coordinate system.

cycle consistency, we calculate the Euclidean norm of $\mathbf{f}_{ij} + \mathbf{f}_{jk} - \mathbf{f}_{ik}$ as the error. The error smaller, the cycle consistency (invariant on deforming path) maintains better. We conduct experiments on a human body rotated in 360 degrees (200 frames) from KillingFusion [54] dataset and a talking head (300 frames) from our captured dataset. In the experiment, we randomly select $1,000$ groups of frames and calculate the mean error on depth points of object surface. Since the topology-aware network is irreversible, we optimize the corresponding positions with fixed network parameters and ADAM optimizer [32]. As a comparison, we also evaluate them on our framework with 6D motion. As Tab. 3 shown, cycle consistency of the whole deformation field among frames is maintained by bijective map $F_{\mathbf{h}}$ quite well, although it might be affected by irreversible topology-aware network.

## 5 Conclusion

We have presented NDR, a new approach for reconstructing the high-fidelity geometry and motions of a dynamic scene from a monocular RGB-D video without any template priors. Other than previous works, NDR integrates observed color and depth into a canonical SDF and radiance field for joint optimization of surface and deformation. For maintaining cycle consistency throughout the whole video, we propose an invertible bijective mapping between observation space and canonical space, which fits perfectly with non-rigid motions. To handle topology change, we employ a topology-aware network to model topology-variant correspondence. On public datasets and our collected dataset, NDR shows a strong empirical performance in modeling different class objects and handling various challenging cases. Negative societal impact and limitation: like many other works with neural implicit representation, our method needs plenty of computation resources and optimization time, which can be a concern for energy resource consumption. We will explore alleviating these in future work.

**Acknowledgements.** This research was partially supported by the National Natural Science Foundation of China (No.62122071, No.62272433), the Fundamental Research Funds for the Central Universities (No. WK3470000021), and Alibaba Group through Alibaba Innovation Research Program (AIR). The opinions, findings, conclusions, and recommendations expressed in this paper are those of the authors and do not necessarily reflect the views of the funding agencies or the government. We thank the authors of OcclusionFussion for sharing the fusion results of several RGB-D sequences. We also thank the authors of BANMo for their suggestions on experimental parameter settings. Special thanks to Prof. Weiwei Xu for providing some help.

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
