# OpenReview forum: "Neural Surface Reconstruction of Dynamic Scenes with Monocular RGB-D Camera"
_NeurIPS.cc/2022/Conference — NeurIPS 2022 Accept_

### Official Review · Reviewer_1XXC · 2022-07-10

**Rating:** 7
**Confidence:** 5
**Soundness:** 4 excellent
**Presentation:** 4 excellent
**Contribution:** 3 good

**Summary:**

The paper tackles the problem of dynamic object reconstruction from a single RGBD video. It solves the problem in a differentiable rendering pipeline and designs a novel 3D warping function that is guaranteed to be invertible. It achieves high-quality reconstruction results on KillingFusion, DeepDeform and iPhone videos.

**Questions:**

- For the implementation of invertible wrapping fields in 155-165, if points in the canonical space happen to have the same $w'$ coordinate, their predicted $R_{uv}$ is constrained to be the same. Similarly if points in the observation space have the same $(u, v)$ coordinates, their predicted $\delta_w$ is constrained to be the same. Does this cause undesirable artifacts, for instance, when dealing with an object containing a flat surface with same $w'$ coordinates?
- Eq(13) is not clearly explained. What is v_p? What does it mean to force points on depth images to be visible from camera?
- Fig. 6: the difference between 6D motion fields and full is not obvious. Consider adding more descriptive captions to highlight the difference or choose the example more carefully.

**Limitations:**

Yes

**Strengths And Weaknesses:**

**Strengths**
- The method is conceptually simple and effective. The authors did well in integrating the best existing solutions (neural surface, canonical hyper-space, and invertible NNs) into a system that not only works well but also stays clean.
- The bijective warping field is an interesting technical improvement over SE(3) fields. it ensures 3D warping functions to be invertible by design.
- The results are high-quality and state-of-the-art.
- The paper is well-written with adequate amount of details. Design choices are also well-motivated.

**Weakness**
- Some details can be clarified. As there is genuine ambiguity between camera motion and object motion, it is worth explaining the camera pose initialization in more detail and analyze the failure modes. When does Robust ICP fail? For instance, does it fail when the object exhibits rotational motion? How robust is the method to inaccurate camera pose initialization?
- Some writing could be improved (also see questions). The technical part of sec. 3.1 is not easy to follow possible due to lack of concise equations in l148-165. It is also not obvious what design choice made it invertible. The key idea of coordinate splitting is only mentioned in l150 and the high-level intuition is not conveyed.

---

> ### Author Response · Authors · 2022-08-02
> **Responses to Reviewer 1XXC**
>
> We thank the reviewer for thinking our method is conceptually simple and effective, obtaining high-quality results. We also thank the reviewer for thinking our bijective map design is interesting and the paper is well-written. We appreciate the reviewer for the detailed comments and constructive advice. Below are our responses to the questions.
>
> #### **Robustness on camera pose initialization**
>
> - Robust ICP may accumulate errors in long sequences with large rotation (especially more than 90 degrees). Therefore, the initial camera poses still contain a large degree of noises in these cases. However, our NDR can still achieve high-quality reconstruction result, as shown in the right example of Fig.3.
> - As reported in `All-Q2`, we add an experiment to evaluate the NDR's robustness against various degrees of noises of camera poses.
> - We will add more analyses to camera pose initialization in the revised version.
>
> #### **Writing of technical part in Sec. 3.1**
>
> Thanks for pointing this out. The cause of the invertible property is that after specifying a certain coordinate axis, each block predicts the movement along and rotation around the axis in turn, and the process of predicting the deformation is reversible. According to Fig.2(b), focusing on the reverse process, starting from $(u',v',w')$, each block can infer the rotation around and movement along the axis and invert them in turn to recover the original $(u,v,w)$. We will add more clear discussions and formulas in the revised version.
>
> #### **Property of invertible wrapping fields**
>
> In fact, this design will not result undesirable artifacts. Each block of bijective map $F\_\textbf{h}$ only represents a motion related to a certain coordinate axis. And the entire network contains multiple blocks, decomposing motion in 3D space into motions related to multiple axes. We have further discussed the design in Sec. A and C.1 of supplementary material. Taking a plane $w=a$ (constant) for example, points on the surface have different predicted motions related to $u$- and $v$-axis at least.
>
> #### **Unclear explanation of Eq.13**
>
> $\textbf{v}\_\textbf{p}$ refers to the view direction in the 3D canonical space, which is $J\_\textbf{p}(\textbf{p}\_i)\textbf{v}\_i$ determined by the view direction $\textbf{v}\_i$ in the observation space and transformed via the Jacobian matrix $J\_\textbf{p}(\textbf{p}\_i)$. The design purpose of visible loss term is to constrain the angle between the normal vector of the sampled point on depth map and the view direction to be larger than 90 degrees, which aims to constrain depth points to be visible surface points under the camera view. We have demonstrated and analyzed this loss in Sec. B of supplementary material. We will modify the related part in the revised version.
>
> #### **Inapparent difference in Fig.6**
>
> Thanks for pointing this out. We will add annotations on results or replace them with other examples in Fig.2 of supplementary material in the revised version.

---

> > ### Author Response · Authors · 2022-08-08
> > **Looking forward your comments on our rebuttal**
> >
> > Dear Reviewer 1XXC,
> >
> > Thank you again for your review. We hope that our rebuttal could address your questions and concerns. As the discussion phase is nearing its end, we would be grateful to hear your feedback and wondered if you might still have any concerns we could address.
> >
> > Thank you for your time.

---

> > > ### Comment · Reviewer_1XXC · 2022-08-08
> > > **Thanks for the rebuttal.**
> > >
> > > My questions are addressed adequately. This is a technically sound paper with impressive results.
> > >
> > > An additional comment on baseline comparison (Fig 5) -- Banmo results look much less smooth than those presented in the paper, containing extruding edges on the human face and symmetric pattens on the cloth. I recommend reaching out to the authors and confirm whether this is expected.

---

> > > > ### Author Response · Authors · 2022-08-09
> > > > **Thanks for the reply!**
> > > >
> > > > Thank you for your quick reply and review comments. For BANMo results shown in Fig.5 of the main paper:
> > > >
> > > > - Before submission, we have contacted the authors and set the suggested experimental parameters, such as the frequency of Fourier Encoding.
> > > >
> > > > - In our experiments, we find the smoothness of results relates to the range of global rigid motion of objects. The BANMo results in Fig.4 of supplementary material show that feature registration may play a more correct role as the angle of camera view changes distinctly.

---

### Official Review · Reviewer_NfVp · 2022-07-11

**Rating:** 7
**Confidence:** 3
**Soundness:** 3 good
**Presentation:** 3 good
**Contribution:** 3 good

**Summary:**

This paper proposed a method for surface reconstruction from a sequential RGBD input. The main strategy for this method is the cycle consistency between canonical and observation space. This approach seems to represent non-rigid deformation. Moreover, to be topology-aware, this paper employed [45]. The results show some comparisons with DynamicFusion[42], OcclusionFusion[58], and so on. Both qualitative and quantitative results show that the method proposed in this paper is better than the methods for the comparison in some parts. The ablation study is also conducted but it is qualitative results only.

**Questions:**

As for the canonical space, how is it decided? Moreover, does the choice, deciding, or learning of canonical space affect the performance?
In a case of a long sequence with big motion like a 360 rotation, does the cycle consistency correctly work?

**Limitations:**

If there are some motions that this method cannot apply, I think it should be mentioned as a limitation of this method.


**Strengths And Weaknesses:**

Strength
- The idea of a cycle consistently between observation and canonical space is reasonable.
- Topology-awareness improves the results but it is basically the previously proposed method.
- The organization and writing are good.

Weakness
- As written in the questions and limitations of this review, I cannot find how to define the canonical space.  Also, I have some concerns in the limitation of this method.

---

> ### Author Response · Authors · 2022-08-02
> **Responses to Reviewer NfVp**
>
> We thank the reviewer for thinking our proposed bijective map is reasonable and the writing is good. We also thank the reviewer for the detailed comments and constructive suggestions. Below are our responses to the questions.
>
> #### **Canonical space**
>
> In our method, we do not explicitly select one frame as the canonical frame. The geometry of each frame is represented as canonical geometry plus deformation of the frame. The canonical geometry and deformations are the variables to be optimized. Meanwhile, the proposed bijective map also regularizes the non-rigid motion. With the well designed representation formulation, fitting term and regularization, the optimization tends to represent the canonical space towards the average shape of all frames. In this way, the non-rigid deformation field can be more easily represented and learned.
>
> #### **Cycle consistency in challenging cases**
>
> - As analyzed in Line 138-147, the cycle consistency of bijective map is strictly maintained, which is independent of data.
> - As reported in `All-Q1`, we evaluate the cycle consistency of whole deformation field (bijective map and topology-aware network). Seq. *Rotated Body* records a human body rotated in 360 degrees (big motion), while Seq. *Talking Head* contains 500 frames (long sequence). The results show that the cycle consistency maintaining of deformation field is not affected by the inreversible topology-aware network overly, owing to the design of bijective map.
>
> #### **Failure cases of motions**
>
> Our method might fail for the inputs with both large and fast movements, such as running. In this case, the RGB image is blurry and the depth map contains a lot of noises, and thus the captured RGB and depth data do not contain enough effective information to guide the reconstruction. Meanwhile, it is quite difficult to get a reasonable camera pose as initialization in this case. We will add the discussions in the revised version.

---

> > ### Comment · Reviewer_NfVp · 2022-08-10
> > **Thank you for the reply**
> >
> > Thank you for addressing my concerns.
> > I modified my rating.
> >
> > best

---

> > > ### Author Response · Authors · 2022-08-10
> > > **Thanks for the reply!**
> > >
> > > Dear Reviewer NfVp,
> > >
> > > Thank you for your quick reply and review comments!
> > >
> > > Best regards, Paper2241 authors

---

> ### Author Response · Authors · 2022-08-08
> **Looking forward your comments on our rebuttal**
>
> Dear Reviewer NfVp,
>
> Thank you again for your review. We hope that our rebuttal could address your questions and concerns. As the discussion phase is nearing its end, we would be grateful to hear your feedback and wondered if you might still have any concerns we could address.
>
> Thank you for your time.

---

### Official Review · Reviewer_1SCx · 2022-07-11

**Rating:** 7
**Confidence:** 4
**Soundness:** 4 excellent
**Presentation:** 3 good
**Contribution:** 3 good

**Summary:**

This paper proposes a template-free RGB-D based 3D scene reconstruction method that handles non-rigid changes with deformation in dynamic scenes.
The proposed method, Neural Dynamic Reconstruction (NDR) follows the similar steps with the classic dynamic scene reconstruction methods.
It uses a neural implicit function for surface representation, and thus using neural signed distance fields (SDF) and proposes a novel neural invertible deformation network that utilizes a bijective map between frames and a canonical space in order to constrain the non-rigid deformation of observed surfaces.
The paper also adds a topology aware network that tackles the well known challenges of dynamic scene reconstruction of the free-form deformation, where dramatic changes of topology (or assumptions of topology) could make handling deformation/motion constraints hard.
The experiments show that the proposed method outperforms (in the context of the accuracy of surface reconstruction over time) the other existing monocular dynamic scene reconstruction methods.


**Questions:**

To make the paper more solid and to help the readers understand the article more clearly, I enumerated several questions below. Some questions may focus on how far the use of the proposed method is from the real world applications.

-How much offset the method can handle refining camera poses, and how much error or residual that wrongly initialized camera pose could be handled?

-How the segmentation results affect the overall results? How do the residuals in the boundary of target surfaces affect the quality of the reconstruction results?

-What is the sole inference time (including optimization steps) for each example? Particularly, providing some numbers, settings (settings, pose accuracy, # samples for each example demonstrated in the result sections would be very helpful to understand the correlation between the complexity of the scene/topology/motion.

-Is there any number or visualization that shows how accurately the bijective map is constructed?

-How wrongly handled topology affects the bijective map quality?

Regarding the importance of the major contribution of this paper, rather than the comparison to the 6D motion, extra discussion on the bijective map and its dependency on the actual topology aware network would make the paper more solid.

-The method is obviously very expensive. Is there any potential idea or open discussion to reduce the computational expenses? For example, not completely evaluate cycle consistency all over the frames, making some steps sequentially updatable. etc.

-In Figure 3. the results from DynamicFusion do not look right. What is the main source of the stripe artifacts in the most of results?

**Limitations:**

As addressed in the conclusion, the major limitation of the method is probably the computational expense.
At least adding a small section about the discussion of how to make the method scalable (even with the trade off between quality) or how to make it easier to use the method in the general use cases (end to end scenario) would make the paper more solid.
Finally, as the major contribution also lies in the use of bijective map and topology aware network, providing more discussion and detailed evaluation (in addition to ablation test) would make the paper even more solid.

**Strengths And Weaknesses:**

I appreciate the research effort from the authors. The results look very impressive and the contribution look very clear. Here are the +/- of the proposed methods and the submitted article.

+Convincing results compared to existing methods

+Provides solutions of each challenging limitation of classic methods (and other latest methods).

+Great idea on the use of bijective map together with topology aware network

-Need more detailed discussion and the evaluation of the bijective map and topology aware network

-Need more detailed discussion of computational expenses and how to handle them.

---

> ### Author Response · Authors · 2022-08-02
> **Responses to Reviewer 1SCx**
>
> We thank the reviewer for thinking our paper has a great idea, provides solutions to some challenges and exhibits convincing results. We appreciate the reviewer for the detailed comments and constructive suggestions which can make the paper more solid. Below are our responses to the questions.
>
> #### **Discussion and evaluation of bijective map and topology-aware network**
>
> As reported in `All-Q1`, we evaluate the effect of bijective map and topology-aware network to cycle consistency maintaining. It shows that the bijective map does help preserve the deforming path independency among arbitrary three timestamps. We will add a visualization of correspondence estimation results and an ablation study on topology-aware network to the revision. The ablation study is to evaluate the quality of estimated correspondence without topology-aware network, especially in topology-variant regions.
>
> #### **Computational expenses and how to handle them**
>
> The following strategies might be useful to improve the computation speed.
> - We might do not need differentiable rendering at the beginning of optimization, which is a time-consuming step. The training strategy is to let deformation field and neural SDF converge to a good initial value only with the captured depth maps.
> - Based on the converged result of item 1, we can directly sample some points near the approximate surface, which also saves sampling time in free-space. Then we leverage RGB information to jointly refine reconstruction.
> - We will consider compact geometry representations (e.g., Instant-NGP [Müller et al. 2022]) and combine with our NDR in the future work. It should be pointed out that our method does not need to completely evaluate cycle consistency all over the frames during optimization, since the cycle consistency of bijective map is preserved naturally.
>
> #### **Robustness on camera pose initialization**
>
> Please refer to the response in `All-Q2`.
>
> #### **The effect of segmentation results**
>
> Like most dynamic neural implicit methods (e.g., BANMo), inaccurate object segmentation will affect the reconstruction results. As shown in the duck toy of Fig.1, when a small region of the background is incorrectly segmented into the foreground, the reconstructed geometry also varies correspondingly in this region. A possible solution is to further optimize the segmentation results in our reconstruction framework based on the segmentation consistency from different views, which could further improve the reconstruction quality.
>
> #### **Inference time and example settings**
>
> The testing videos have at least 200 frames and at most 500 frames, which is related with the motion complexity. The optimization time of a single scene on a single A100 GPU is about 12 hours, which has been discussed in Line 228. We will add related experimental settings to the revision.
>
> #### **Stripe artifacts in DynamicFusion results**
>
> As shown in Fig.3, it mainly depends on the degree of global rotation around the $y$-axis and the number of deformation nodes, which affect the interpolation continuity of rotation of vertices. On the other hand, it is also relevant to the smoothness of the dependence weight of vertices on deformation nodes. The stripe artifacts tend to disappear when the object rotates wildly (shown in 2:00-2:12 of demo video) or indistinctively (shown in Fig.4).
>
> [Müller et al. 2022] Thomas Müller, Alex Evans, Christoph Schied, and Alexander Keller. Instant neural graphics primitives with a multiresolution hash encoding. *ACM Transactions on Graphics (TOG)*, 41(4):1–15, 2022.

---

> ### Author Response · Authors · 2022-08-08
> **Looking forward your comments on our rebuttal**
>
> Dear Reviewer 1SCx,
>
> Thank you again for your review. We hope that our rebuttal could address your questions and concerns. As the discussion phase is nearing its end, we would be grateful to hear your feedback and wondered if you might still have any concerns we could address.
>
> Thank you for your time.

---

### Official Review · Reviewer_BjMr · 2022-07-17

**Rating:** 6
**Confidence:** 3
**Soundness:** 3 good
**Presentation:** 3 good
**Contribution:** 3 good

**Summary:**

This paper introduces a template-free method to reconstruct a high quality geometry and motion of a dynamic scene from a single RGB-D camera. It proposes a bijective deformation map to preserve the cycle consistency between two frames, thus it doesn’t require any scene or optical flow map. To handle the topology changes, the deformation network is combined with a topological-aware network. Experimental results show that the proposed method overcomes the state-of-the-art RGBD methods, such as DynamicFusion and OcclusionFusion, and RGB methods, such as BANMO.

**Questions:**

- While the proposed method utilizes depth from Kinect as the ground truth, is there any effect if the captured depth information is noisy? Note that an active depth camera could not be used as the ground truth due to its noisy characteristic?
- Why don’t use a synthetic dataset for the ablation study to prove the proposed idea?


**Limitations:**

The authors have addressed the limitations in the conclusion.

**Strengths And Weaknesses:**

# Strenghts

- A novel 3D reconstruction network of a dynamic scene from a single RGB-D camera. The combination between topology-aware network and deformation-based network makes the network to model the geometry and motion of a dynamic object.
- The proposed bijective map is able to preserve the cycle-consistency because it models the points in 3D observation space to the points in 3D canonical space.

# Weaknesses

- There is only a qualitative ablation study. The overall framework is like an extended version of HyperNERF for RGBD videos. Thus, it is essential to perform a quantitative ablation study, especially compared to HyperNERF.
- Lack of quantitative experimental results. The proposed method only performs a qualitative evaluation for various comparison methods. While qualitative evaluation can be subjective, it is essential to perform quantitative evaluation, especially with BANMo, HyperNERF, VolumeDeform, etc.
- It is unclear about the cycle-consistency performance of the proposed bijective map. The evaluation method only focuses on a single frame. There should be a way to evaluate the consistency between frames because it is also a part of the proposed contribution.
- It is recommended to follow the HyperNERF Fig. 8 to show the performance of the proposed method. It is unclear how the topology-aware and bijective map-based deformation networks affect the overall performance.

---

> ### Author Response · Authors · 2022-08-02
> **Responses to Reviewer BjMr**
>
> We appreciate the reviewer thinks our reconstruction network is novel. We also thank the reviewer for the detailed comments and constructive suggestions. Below are our responses to the questions.
>
> #### **Qualitative ablation study and relationship with HyperNeRF**
>
> - Except ablation studies in the paper, some other qualitative ablation studies are presented in the supplementary material, including Fig.2 and 2:40 to 3:06 of demo video. The effect of each component has been clearly presented and analyzed.
> - HyperNeRF is a method for novel-view synthesis which does not include a reconstruction module, such as SDF or Occupancy representation. The difference of our method with HyperNeRF is similar with NeuS and NeRF. Except different targets, we also propose a new representation with bijective mapping and several terms to utilize captured depth, which are all specifically designed for our task.
> - Actually, the baseline with 6D motion in ablation study can be treated as an RGB-D dynamic reconstruction version of HyperNeRF. As shown in Fig.6 of paper and Fig.2 of supplementary material, our NDR can not only preserve geometry details but also reduce artifacts.
> - As reported in `All-Q1`, we add a quantitative evaluation on cycle consistency, which also compares the baseline with 6D motion. It can be seen that our bijective map does help preserve cycle consistency.
>
> #### **Lack of quantitative experimental results**
>
> We have tried our best to do comparisons with existing methods. We have compared DynamicFusion and OcclusionFusion on 2 common datasets and our captured dataset on the most common geometry metric to evaluate reconstruction quality. For some other methods without public codes and results, we also did qualitative comparisons (Fig.4).
> - From the qualitative experiments, we can see that our NDR outperforms existing methods by a large margin, which have fully demonstrated the superiority of our method.
> - We also conduct quantitative comparisons between our NDR with other methods (Tab.1). As described in Sec. C.1 of supplementary material, some methods do not release their codes and results. This makes it difficult for us to conduct quantitative experimental comparisons with them.
> - For BANMo: Fig.5 could clearly show that our full model outperforms BANMo. We can add a quantitative evaluation: register reconstructed model of BANMo to depth frame and calculate the metric.
> - For HyperNeRF: As stated above, the baseline with 6D motion can be seen as a comparison with RGB-D reconstruction version of HyperNeRF.
> - For VolumeDeform: We have quantitatively compared a state-of-the-art RGB-D method, OcclusionFusion (CVPR 2022). On the other hand, as far as we know, the code of VolumeDeform is not public available.
>
> #### **Unclarity about cycle-consistency performance**
>
> We have analyzed the strict cycle-consistency of bijective map in Line 138-147. The evaluation of whole deformation field is reported in `All-Q1`.
>
> #### **Fig.8 of HyperNeRF**
>
> We will add an ablation study on topology-aware network and follow HyperNeRF's Fig.8 exhibition (including geometry under novel views) to the revision.
>
> #### **Effects of captured depth noises**
>
> - The noises contained in captured depth map will definitely affect reconstruction quality. To alleviate the effect of noise on modeling quality, we first perform a depth denoising. The denoised depth map provides a reliable geometric constraint for reconstruction, as evaluated in Fig.5.
> - Although a single-frame depth map contains noises, the fusion of depth maps can restore high-quality 3D geometry, which is the key idea of KinectFusion and DynamicFusion. Except for the global fusion of depth maps, our NDR also utilizes image information in a differentiable rendering framework to jointly optimize geometry, thus achieving better reconstruction results. It can be seen in Fig.6 of paper and Fig.2 of supplementary material that the reconstruction quality gets improved with the help of color images.
> - It is true that depth map can not be treated as GT. However, considering that it is very difficult to obtain accurate geometry shapes of non-rigid deforming objects in the reconstruction problem, most of existing geometric reconstruction works (e.g., DeepDeform and OcclusionFusion) adopt denoised depth map as GT.
>
> #### **Synthetic dataset**
>
> Thanks for proposing to use synthetic data for ablation study. We will add this kind of ablation studies to the revision. We think it would be both useful to analyze the effect of each component with synthetic data and real data. As shown in Fig.2 of supplementary material and 2:40 to 3:06 of demo video, we can clearly see the different role of each component on real captured data. In addition, as reported in `All-Q2`, we evaluate the robustness to camera pose initialization on synthesis data from AMA dataset (render meshes to obtain depth frames). It can be seen that the reconstruction quality is more related to the degree of non-rigid motion of sequence.

---

> ### Author Response · Authors · 2022-08-08
> **Looking forward your comments on our rebuttal**
>
> Dear Reviewer BjMr,
>
> Thank you again for your review. We hope that our rebuttal could address your questions and concerns. As the discussion phase is nearing its end, we would be grateful to hear your feedback and wondered if you might still have any concerns we could address.
>
> Thank you for your time.

---

> > ### Comment · Reviewer_BjMr · 2022-08-08
> > **Comments**
> >
> > Dear Authors,
> >
> > I have no additional comments. Your rebuttal has addressed my concerns.
> >
> > Sincerely,

---

> > > ### Author Response · Authors · 2022-08-08
> > > **Thanks for the reply!**
> > >
> > > Dear Reviewer BjMr,
> > >
> > > Thank you for your quick reply and review comments!
> > >
> > > Best regards,
> > > Paper2241 authors

---

### Author Response · Authors · 2022-08-02
**Responses to common concerns**

All reviewers appreciate that our proposed NDR is effective and the results are impressive. We thank the reviewers for their constructive comments. Below are our responses to those common questions.

#### **All-Q1. Evaluation of Cycle Consistency**

We add a numerical experiment for cycle consistency evaluation on the whole deformation field. In the experiment, we randomly select 3 frames (indexed by $i,j,k$) as a group in a video sequence. Given points on one frame, we calculate the corresponding coordinates on another frame, and record this scene flow as $\mathbf{f}$. Then it includes 2 deforming paths from frame $i$ to $k$, based on the direct flow $\mathbf{f}\_{ik}$, or the composite flow $\mathbf{f}\_{ij}+\mathbf{f}\_{jk}$. To evaluate the cycle consistency, we calculate the Euclidean norm of $\mathbf{f}\_{ij}+\mathbf{f}\_{jk}-\mathbf{f}\_{ik}$ as the error. The error smaller, the cycle consistency (invariant on deforming path) maintains better. We conduct experiments on a human body rotated in 360 degrees from KillingFusion dataset and a talking head from our captured dataset. In the experiment, we randomly select 100 groups of frames and calculate the mean error on depth points of the object. Since the topology-aware network is inreversible, we optimize the corresponding coordinates with fixed network parameters and Adam optimizer. As a comparison, we also evaluate them on our framework with 6D motion:

||w/ Bijective Map|w/ 6D Motion|
|:----:|:----:|:----:|
|Rotated Body|5.37|441.12|
|Talking Head|4.91|261.58|

Each value is the mean error ( $\times 10^{-4}$ ) per point in the unit coordinate system. The quantitative results show that the cycle consistency of whole deformation field among frames is maintained by our proposed bijective map quite well, although it might be affected by the inreversible topology-aware network.

#### **All-Q2. Robustness to camera pose initialization**

- For examples of small rigid motion (e.g., Fig.4 and Fig.6), our NDR can achieve high-quality reconstruction results even without Robust ICP based camera pose initialization.
- For examples of large rigid and non-rigid motions (e.g., 2:01 to 2:10 of demo video), the initial poses are not accurate enough although the Robust ICP is adopted. However, our NDR can still refine the camera poses well and obtain impressive reconstruction results.
- In order to systematically analyze the performance of our camera pose optimization module, we add an experiment to test NDR's robustness under various degrees of noises on both real and synthetic dataset. We choose 2 sequences of small rigid motion separately from DeepDeform dataset (a body with moving joints, 200 frames) and Articulated Mesh Animation (AMA) dataset [Vlasic et al. 2008] (a Samba dancer, 175 frames). AMA dataset is a multi-view dataset which contains reconstructed mesh corresponding to each video frame. To construct synthesis depth data, we render meshes to a chosen camera view. In the experiment, we do not utilize any multi-view messages but only monocular RGB-D frames. We add Gaussian Noises with $5,10,20,40,60$ degrees of standard deviation to initial Euler angles and calculate mean geometry errors ( $0$ denotes without adding noises as a reference):

||0|5|10|20|40|60|
|:----:|:----:|:----:|:----:|:----:|:----:|:----:|
|Moving Joints|2.95|4.58|6.58|9.17|11.07|30.28|
|Samba Dancer|3.90|5.49|8.18|11.60|13.46|27.02|

All values are in *mm*. The results show that our NDR is robust against noisy camera poses to a certain extent, owing to its neural implicit representation and abundant optimization with RGB-D messages. If the standard deviation of Gaussian Noises is over 20 degrees, the reconstruction quality will be obviously affected (geometry error is over 1 *cm*).

[Vlasic et al. 2008] Daniel Vlasic, Ilya Baran, Wojciech Matusik, and Jovan Popović. Articulated mesh animation from multi-view silhouettes. In *ACM SIGGRAPH 2008 Papers*, pages 1-9. 2008.

---

### Meta-Review · Area_Chair_SQTM · 2022-08-31

**Recommendation:** Accept
**Confidence:** Certain

**Metareview:**

This paper had consistently positive reviews from all reviewers and weaknesses that were expressed were responded to coherently by the authors.  I recommend this paper be accepted.

**Award:**

Yes

---

### Decision · Program_Chairs · 2022-09-14

Accept